# Prenatal Detection and Conservative Management of Uterine Scar Dehiscence in Patient with Previous Uterine Rupture and Multiple Surgeries—A Case Report

**DOI:** 10.3390/healthcare12100988

**Published:** 2024-05-10

**Authors:** Silvia Zermano, Giuseppina Seminara, Nadia Parisi, Valentina Serantoni, Martina Arcieri, Anna Biasioli, Monica Della Martina, Stefano Restaino, Giuseppe Vizzielli, Lorenza Driul

**Affiliations:** 1Department of Maternal and Child Health, “Santa Maria della Misericordia” University Hospital, Azienda Sanitaria Universitaria Friuli Centrale (ASUFC), 33100 Udine, Italy; martina.arcieri@asufc.sanita.fvg.it (M.A.); anna.biasioli@asufc.sanita.fvg.it (A.B.); monica.dellamartina@asufc.sanita.fvg.it (M.D.M.); stefano.restaino@asufc.sanita.fvg.it (S.R.); giuseppe.vizzielli@uniud.it (G.V.); lorenza.driul@uniud.it (L.D.); 2Department of Medicine (DAME), Università degli Studi di Udine, Via delle Scienze, 206, 33100 Udine, Italy; seminara.giuseppina@spes.uniud.it (G.S.); serantoni.valentina@spes.uniud.it (V.S.); 3Obstetric-Gynecologic Unit, General Regional Hospital “F. Miulli”, Acquaviva delle Fonti, 70021 Bari, Italy; n.parisi@miulli.it

**Keywords:** uterine dehiscence, uterine rupture, adenomyosis, ultrasound detection of uterine dehiscence, cesarean delivery

## Abstract

Uterine rupture is a rare and life-threatening condition. It usually occurs in patients with uterine scars (most commonly for a previous myomectomy or caesarean section), but it can also affect an unharmed uterus. This complication is more frequent in the third trimester and during delivery. There is not yet a recognised method of prediction of uterine rupture and the ultrasound features still need a consensus. In this article, we have reported a case of uterine dehiscence diagnosed by a pelvic ultrasound and magnetic resonance (MRI) at 24 weeks of gestation. The finding was confirmed intraoperatively at the caesarean section at 29 weeks of gestation. The 40-year-old patient has had a previous pregnancy complicated by uterine rupture at 22 weeks of gestation, following six previous abdominal surgeries for stage IV endometriosis, diffuse and nodular adenomyosis, and pelvic adhesion syndrome. The early detection of uterine dehiscence allowed us to prolong the pregnancy and perform a subsequent fertility-sparing surgery, reducing maternal and neonatal morbidity and mortality. Our case report proves that women with severe endometriosis/adenomyosis are at a high risk of uterine rupture and scar dehiscence. The antenatal ultrasound can describe a uterine dehiscence (even in asymptomatic patients) and prevent complications.

## 1. Introduction

The incidence of caesarean delivery is increasing worldwide, as is the incidence of complications that can result. [1] Uterine rupture is a rare event, defined as a spontaneous complete tear within the uterine wall, occurring during or before labor. The outer uterine lining, serosa, is damaged, with possible bleeding and fetal displacement in the abdominal cavity. Differently, in uterine dehiscence, the visceral peritoneum is intact and the fetus remains in the uterine cavity [2,3,4]. For this reason, uterine dehiscence is most often asymptomatic and diagnosed as an incidental finding of a repeated caesarean section at term. An undetected uterine dehiscence may occur much earlier, putting the patient at a high risk of uterine rupture [5,6]. Although the incidence of uterine dehiscence between the end of the second and beginning of the third trimester is extremely low, it can be a potentially catastrophic complication in pregnancy [7]. The uterine rupture rate in Italy is 0.16/1000 pregnancies, according to data published in 2021 by Donati and colleagues [8]. Uterine rupture can occur on an unscarred uterus or after a caesarean section, a myomectomy, or an adenomyoma excision, or from a hysteroscopic surgery of congenital uterine malformations [2,8,9]. The uterine rupture risk factors are as follows: a previous uterine rupture (risk of recurrence is 6%), a previous fundic hysterotomy, a T or J previous hysterotomy (risk is 4–9%), the induction of labour (especially with prostaglandins, that increase the risk of 2.45%) [10], a previous low vertical section (1–2%), two previous caesarean sections (the risk is 1.59%), one previous low transverse section (the risk is 0.4–0.7%), and spontaneous labour. Other risk factors are as follows: gestational age over 40 weeks, maternal age of 40 years or more, obesity, lower prelabour Bishop score, fetal macrosomia, decreased ultrasonographic lower segment myometrial thickness, and a short interval between the births [11].

In 2021, a research showed 14 cases out of 74 pregnancies with uterine rupture between 22–31 weeks of gestation, confirming the highest frequency in the third trimester (mean gestational age was 35 weeks) of pregnancy. Additionally, preterm deliveries showed an incidence of uterine rupture higher by more than three times in comparison with term delivery [8]. This case report describes an ultrasound and MRI diagnosis of uterine dehiscence performed at 24 weeks of gestation. The complication was then confirmed intraoperatively at the time of caesarean section at 29 weeks of gestation. The patient had a previous uterine rupture at 22 weeks and multiple surgeries for severe endometriosis and adenomyosis. Showing this case, we would suggest the early screening for uterine dehiscence in order to prevent maternal and neonatal morbidity and mortality. This could be very clinically relevant especially in patients with a compromised abdomino-pelvic situation due to grade IV endometriosis, uterine adenomyosis, previous pelviperitonitis, and multiple previous surgeries.

## 2. Case Report 

A 40-year-old G2P1A0L0 woman, with a previous uterine rupture at 22 weeks, was admitted for observation to our Obstetrics Department at 24/5 weeks after a routine ultrasound scan with the suspicion of uterine dehiscence. Magnetic resonance showed an endometriotic lesion on the previous uterine scar and a doubtful dehiscence area. 

The patient was completely asymptomatic. Her vital signs and laboratory tests were within the normal range and the pelvic examination showed a closed and long cervix. Fetal heart rate tracing was normal, and no uterine contractions were detected on tocometry. 

The patient’s history showed multiple abdominal surgeries for treatment of severe endometriosis-adenomyosis and abdomino-pelvic adhesions. (Figure 1) 

In July 2016, she suffered from pelviperitonitis and haemoperitoneum due to the rupture of a right ovarian endometrioma, which was removed in laparotomy with the lysis of numerous bowel adhesions. Subsequent imaging diagnostic tests revealed diffuse and nodular adenomyosis, a left ovarian endometrioma, a suspected haemosactosalpinx, and diffuse endometriosis stage IV. 

In November 2016, in a different hospital, the patient underwent laparoscopic surgery with the eradication of deep pelvic endometriosis, a bilateral ureterolysis with bladder shaving, the excision of the uterine adenomyoma, an ileocecal resection, and the resection of the rectosigma with a nerve-sparing technique (Figure 2 and Figure 3). A chromosalpingoscopy is also performed, which highlighted the bilateral tubal patency.

In 2017, following the development of a large median laparocele, the patient underwent laparotomy for surgical repair.

Two years later, the patient was pregnant for the first time. She was admitted for aggravating abdominal pain at 22 weeks of gestation. At that time, we diagnosed a uterine rupture with intrauterine fetal demise. During surgery, an anterofundal uterine wall rupture was seen with haemoperitoneum and the fetus moved into the abdominal cavity. A triple-layer suture with detached stitches was performed to repair and reconstruct the uterine wall.

At discharge, the patient was fully informed of the risks and possible complications of uterine rupture and a report with the available literature data was given. Given the high risk of a recurrent uterine, a second pregnancy was strongly discouraged. 

After several failed attempts to conceive, the patient underwent operative laparoscopy for the lysis of several abdomino-pelvic adhesions, and, in June 2021, laparoscopic urological surgery for the treatment of a severe right ureteral stenosis.

Given the patient’s compromised medical history, the patient was advised against a new Assisted Reproductive Technology (ART) procedure, however. Despite this and being aware of the risks, the couple decided for a Level II ART procedure, which was performed abroad and resulted in the current pregnancy. 

The patient has been strictly monitored since the beginning of the pregnancy in our antenatal clinic. Her pregnancy started without complications. The anatomy fetal scan at 24/5 weeks suspected a uterine wall defect in the lower uterine segment at the level of cervical junction, likely at the site of her previous hysterotomy. The patient was admitted and an additional scan was done. Ultrasound images showed a scar dehiscence overlying the placenta. A placenta accreta was ruled out for the absence of retroplacental hypervascularity, bridging vessels, bulging of the lower uterine wall, and a normal bladder serosa–uterine wall interface (Figure 4 and Figure 5).

Magnetic resonance imaging confirmed a dehiscence at least of 3 cm, adjacent to the suspect endometriotic foci involving the lower uterine segment serosa and a normal-appearing bladder; there was no abdominal free fluid (Figure 6 and Figure 7).

During the admission, due to the onset of contractions at 27/5 weeks, the patient was given two doses of betamethasone 12 mg intramuscularly for fetal lung maturity, tocolytic therapy for a period of 48 h, and neuroprophylaxis with magnesium sulphate at 29/5 weeks as a prevention in the event of childbirth.

At 28 weeks, we repeated the ultrasound assessment and pelvic MRI that showed a slight increase in the wall defect, which reached the size of 4 × 4.5 cm.

Due to persistent uterine contractions and her previous surgical history, a caesarean section was performed at 29/1 weeks of gestation. A large subserosal dehiscence with many blood vessels under the serosa surface, and strong adhesions between the bowel and lower uterine segment were intraoperatively noted. For these reasons, we performed a transverse uterine fundal incision. A vital male baby weighing 1340 g was delivered and the placenta was easily removed. After adhesiolysis, the fundal incision and the dehiscence site were repaired using double-layer closure in Vicryl. Subsequently, we used one vial of 5 mL FloSeal© on breech for haemostasis. Hysterectomy was not required (Figure 8 and Figure 9).

At the end of the surgery, the total volume of blood loss was 3700 mL, and the patient was transfused with two units of red blood cells. The postoperative period was uneventful. Moreover, we used a PICO^TM^ system (Smith & Nephew Medical Ltd., 101 Hessle Road—HU3 2BN, Hull, UK) for a negative-pressure wound therapy across the entire dressing to the wound or incision and peri-wound, while simultaneously removing the exudate. The patient was discharged 5 days after surgery.

## 3. Discussion

An accurate assessment of the uterine wall may be required for these patients with asymptomatic uterine dehiscence, especially those who had a previous history of uterine rupture, caesarean sections, and surgery for endometriosis/adenomyosis [12].

Currently, there is not an established method to predict imminent uterine rupture. Symptoms like pelvic discomfort, abdominal pain, or abnormal foetal heart frequency may be connected to imminent uterine rupture. However, the presence of symptoms alone cannot exclude the possibility of the successful expectant management. In addition, there is still no consensus on the cut-off values for the myometrium thickness for the prediction of spontaneous uterine rupture. The most recent systematic review and meta-analysis suggested that a lower uterine segment (LUS) > 3.65 mm should be safe for a Vaginal Birth After Cesarean Section (VBAC), 2–3.65 mm is probably safe, and <2 mm identifies a patient at a higher risk for uterine rupture/dehiscence. However, the heterogeneity of the considered studies uncertainly defined the prediction of uterine rupture risk based on the thickness of the lower uterine segment [13,14].

Furthermore, a consensus is still missing on the optimal delivery timing in the case of an ultrasound suspicion of a uterine scar dehiscence, especially in the case of a diagnosis in the second or early third trimester of pregnancy.

Previous uterine surgery is a well-known risk factor for uterine rupture even before labor, as previously described. Furthermore, some authors describe that previous myomectomy/adenomectomy can induce a tight intestinal adhesion at the site and may mask the symptoms and signs of a uterine rupture. Although it could not be determined whether intestinal adhesion delayed the diagnosis of a rupture, we must consider this possibility in pregnant women after myomectomy. Moreover, intestinal adhesion occurs not only after myomectomy but also after any other abdominal surgeries, and, thus, in treating our patient with a history of multiple abdominal surgeries, we were even more concerned about this possibility [15].

Since uterine dehiscence has the potential risk for a complete uterine rupture, and acute life-threatening complications for both mother and baby, it is difficult to determine whether to manage expectantly or surgically, including the termination of pregnancy or surgical repair of the uterine wall, especially in the early second trimester. Factors to consider in the decision-making process of preterm delivery include gestational age and the risks of developmental sequelae of prematurity, foetal demise, uterine rupture, and preterm labour [16]. There are few case reports demonstrating that, even if uterine dehiscence was diagnosed by ultrasound in the second trimester, a conservative management through close observation was possible. However, in some of these cases, the diagnosis of dehiscence was only made intraoperatively and ultrasound data were retrospectively collected [17,18,19].

An elective caesarean section before the onset of labour is the best strategy to prevent neonatal and maternal morbidity and mortality. If delivery is planned before 37 weeks of gestation, prenatal steroid administration should be considered to reduce the risk of neonatal respiratory distress [16]. In our case, a caesarean section was scheduled at 29 weeks because of previous uterine rupture and scar dehiscence, but the timing was determined in consideration of the persistence of uterine contractions despite tocolytic therapy.

In conclusion, the emerging evidence is exploring the relationship between endometriosis/adenomyosis and obstetrical complications. It is well-known that a uterine rupture is more frequently related to a scarred uterus or the presence of a minor resistance area in the uterine wall caused by the surgical excision of endometriotic lesions or adenomyomas. However, the evidence is not available as the frequency of these events may have been underestimated due to unreported cases. Roberti Maggiore and colleagues in their literature review confirmed that there is a greater risk of uterine rupture in women with a histological diagnosis of endometriosis or adenomyosis; this group showed that the treatment of the endometriosis of the uterine isthmus or an excision of rectovaginal nodules increased the risk of rupture during pregnancy or delivery. The absence of strong evidence on this issue does not support any form of prophylactic surgery [20]. Iemura and colleagues reported that, in women with diffuse uterine leiomyomatosis and adenomyosis in pregnancy, the risk of a silent uterine rupture with a herniated amniotic sac may increase due to a reduction in the ability to stretch the uterine wall [21].

## 4. Conclusions 

To answer the clinical question, a literature search was performed using the PubMed database (www.ncbi.nlm.nih.gov, accessed 22 April 2024). The search terms were ‘uterine dehiscence’ AND ‘previous surgery’. We found 237 results. Articles describing one or more cases of uterine dehiscence in pregnancy published between 1964 and 2024 were identified from the above database. Only articles written in English were included in our search. Studies describing at least one case of uterine dehiscence in pregnancy in a uterus with previous surgical scarring were considered eligible for inclusion in this review. Any duplicate studies were excluded. The selected articles were independently reviewed by two authors (G.S. and S.Z.). We selected 15 articles from which we collected the following information: number of patients, type and number of previous surgeries, gestational age at diagnosis and at delivery, conservative approach, previous uterine rupture, diagnosis of endometriosis or adenomyosis, fetal exitus and availability of ultrasound, and surgical or MRI images. All these data are listed in Table 1 and compared with the data from our work. This analysis shows that only 17 patients were diagnosed prior to surgery, and, of these, only 10 were managed conservatively. Few patients had a history of previous uterine rupture and only one had a previous diagnosis of endometriosis.

Our work is also the only one to include a graphic documentation of the patient’s previous surgeries and images of the ultrasound, radiological, and surgical diagnoses.

In light of the available data, we can state that further research is needed to determine whether the implementation of a universal ultrasound screening of the lower uterine segment for scar dehiscence in patients with a significant risk of uterine rupture in the second and early third trimester will help to prevent catastrophic events that may follow uterine rupture.

Our case suggests that antenatal ultrasound can lead to a suspicion of uterine dehiscence, even in asymptomatic patients. Another strength of our clinic case is the complexity of the patient’s past medical history, suggesting that new pregnancies for patients with previous uterine rupture, particularly those with severe endometriosis/adenomyosis, are at very high risk, in particular, the risk of uterine scar dehiscence and repeated uterine rupture, with all the consequences that this entails. 

In our clinic, physicians involved in the care of pregnant patients must be aware of the risks associated with previous abdominal surgery, especially those related to endometriosis/adenomyosis, possible complications during pregnancy, and methods of evaluation and treatment. Obstetricians play a central role in assessing the presence of possible complications at the time of ultrasound investigations. The assessment and management of these patients must be the main concern of the health care team, because they affect the survival of the mother and the fetus. The assessment and monitoring must be determined and standardized according to gestational age.

It would be interesting to assess with epidemiological cohort studies the real incidence of obstetric complications in patients with a history of endometriosis/severe adenomyosis and multiple previous surgeries as we reported with this case.

## Figures and Tables

**Figure 1 healthcare-12-00988-f001:**
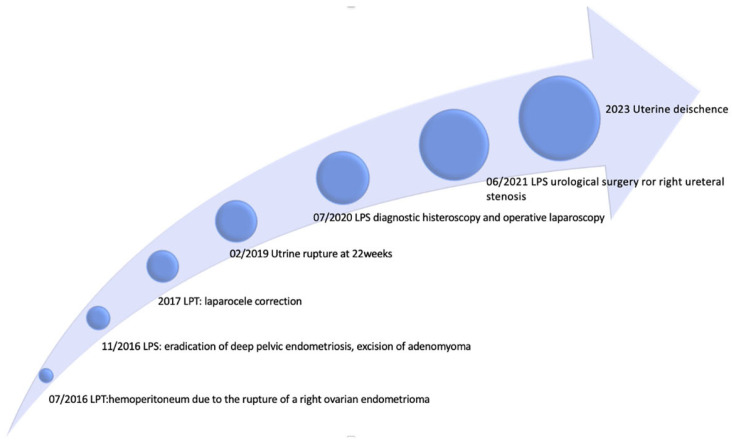
Patient’s surgical history timeline.

**Figure 2 healthcare-12-00988-f002:**
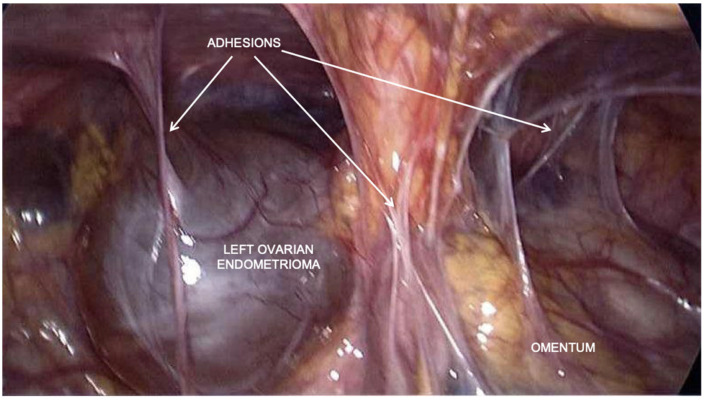
Laparoscopic view before surgery: presence of adhesions and left ovarian endometrioma during laparoscopic eradication surgery in November 2016.

**Figure 3 healthcare-12-00988-f003:**
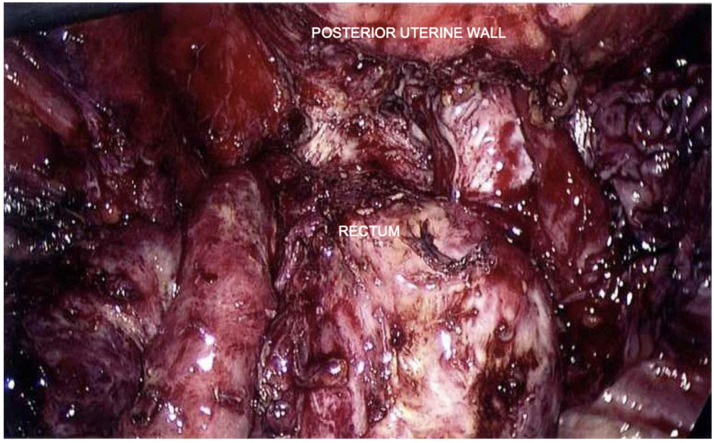
Laparoscopic view after surgery: eradication of deep pelvic endometriosis, bilateral ureterolysis, bladder shaving, excision of uterine adenomyoma, and segmental bowel resection using nerve-sparing technique.

**Figure 4 healthcare-12-00988-f004:**
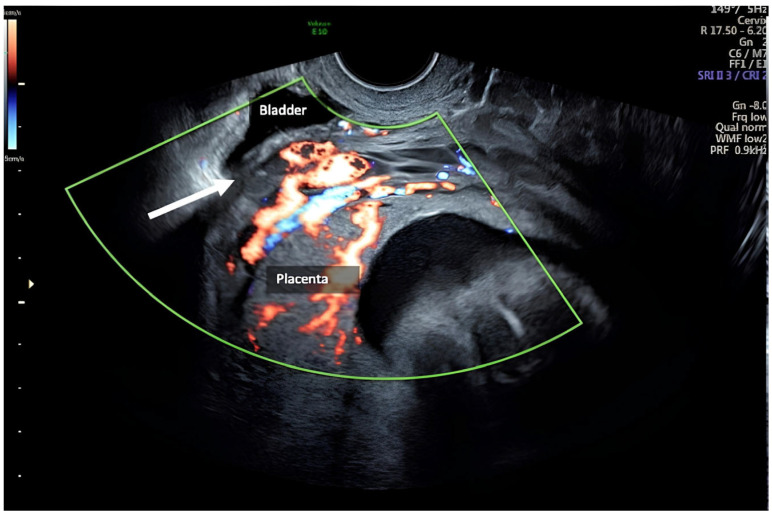
The placenta: scar dehiscence overlying the placenta, absence of retroplacental hypervascularity, bridging vessels, bulging of the lower uterine wall, and a normal bladder serosa–uterine wall interface.

**Figure 5 healthcare-12-00988-f005:**
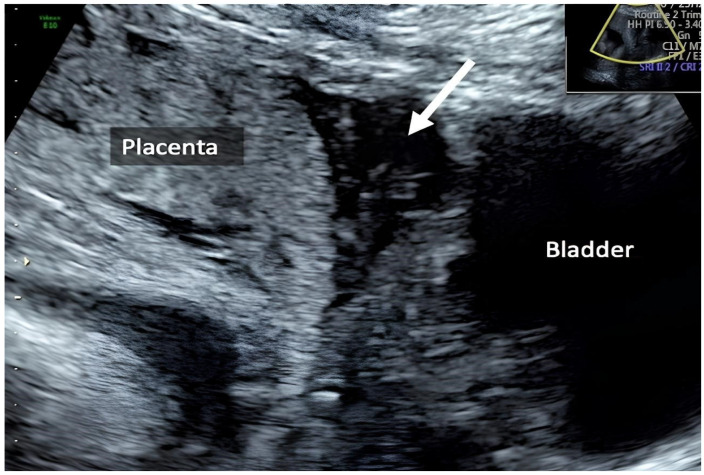
The myometrium: The 24-week scan showing porous consistency of myometrium in the lower uterine segment.

**Figure 6 healthcare-12-00988-f006:**
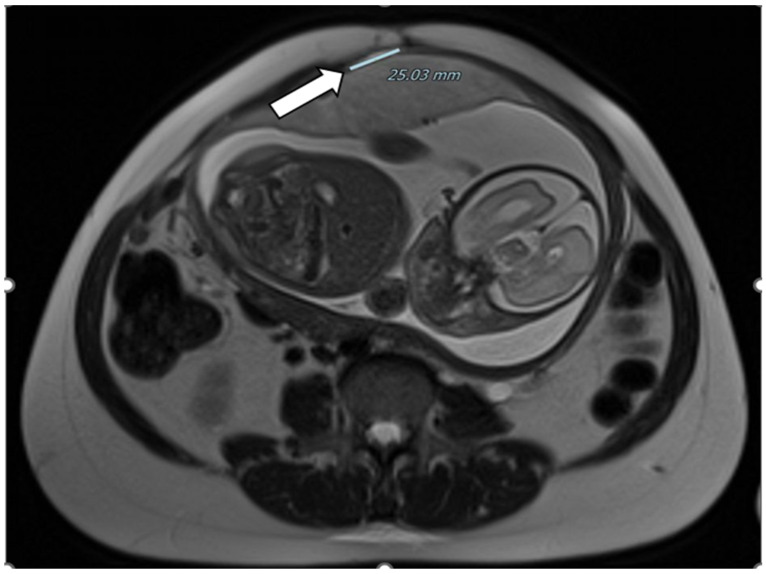
Dehiscence at MRI: presence of a dehiscence adjacent to suspect endometriotic foci involving the lower uterine segment serosa and a normal-appearing bladder.

**Figure 7 healthcare-12-00988-f007:**
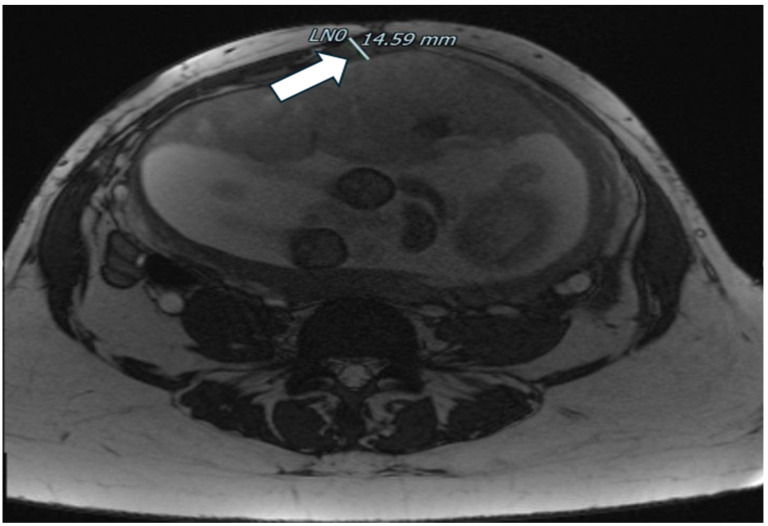
Dehiscence at MRI: another MRI image of dehiscence.

**Figure 8 healthcare-12-00988-f008:**
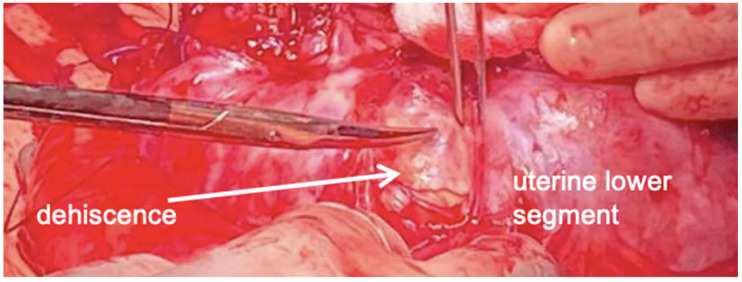
A large subserosal dehiscence with many blood vessels under the serosa surface, and strong adhesions between bowel and lower uterine segment were intraoperatively noted.

**Figure 9 healthcare-12-00988-f009:**
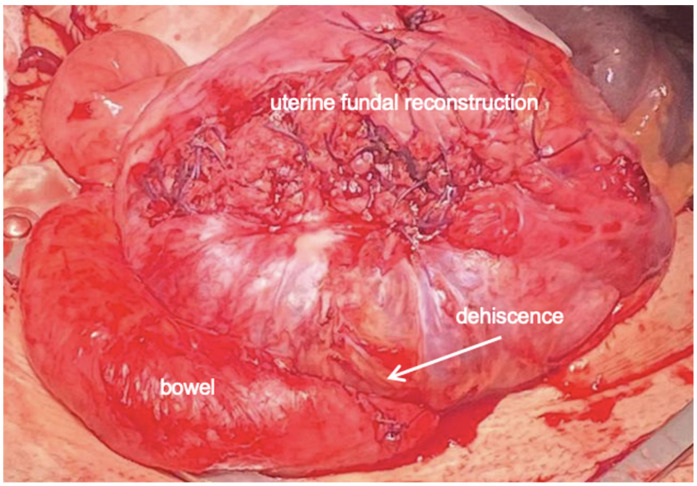
Transverse uterine fundal incision was repaired using double-layer closure in Vicryl.

**Table 1 healthcare-12-00988-t001:** A comparison between our paper and the literature.

Author	Year	PMID	Type	Number of Patients	Number of Previous Surgery	Type of Previous Surgery	
Zermano S. et al.	2024		Case report	1	6	LPT for pelviperitonitis and hemoperitoneum, LPS with eradication of deep pelvic endometriosis, bilateral ureterolysis with bladder shaving, excision of the uterine adenomyoma ileocecal resection and resection of the rectosigma with nerve-sparing technique; LPT for surgical repair of laparocele; LPT for uterine rupture, LPS for lysis of several abdomino-pelvic adhesions.	
Cui X. et al. [22]	2020	32756164	Retrospective case-control study	107	1	Cesarean section	
Edwards D. et al. [23]	2023	35786305	Video article	1	1	Cesarean section	
Eleje G.U. et al. [24]	2023	37009324	Case report	1	1	Cesarean section	
Mutiso SK. et al. [25]	2024	38183151	Case report	1	4	4 cesarean section	
Zhang J. et al. [26]	2014	25864270	Case report	1	1	Cesarean section	
Wye D. et al. [27]	2014	28191206	Case report	1	3	Interstitial pregnancy treated by laparoscopic, Myomectomy laparotomy	
Hamar B.D. et al. [28]	2003	14607034	Case report	1	1	Cesarean section	
Zhu Y. et al. [17]	2021	34309465	Retrospective review	6	1–2	Cesarean section	
Fox NS. et al. [29]	2014	24785605	Retrospective study	44	1–2	Cesarean section	
Tischner I. et al. [30]	2010	20226423	Case report	1	1	Laparoscopically repaired uterine perforation	
Filipcikovaa R. et al. [31]	2014	23446209	Case series	4	1–4	Cesarean section	
Sbarra M.et al. [32]	2009	19482275	Case series	3	1	Cesarean section	
Balachandran Nair D. et al. [33]	2016	27358091	Case report	1	2	Cesarean section	
Inovay J. et al. [34]	1999	10527971	Case report	1	1	LPS ovarian wedge resection and bilateral salpingectomy	
Zuckerwise L.C. et al. [35]	2011	21768866	Case report	1	2	Cesarean section	
**Author**	**Year**	**PMID**	**Previous uterine rupture**	**Endometriosis**	**Adenomyosis**	**Gestational weeks at diagnosis**	
Zermano S. et al.	2024		Yes	Yes	Yes	24 5/7	
Cui X. et al. [22]	2020	32756164	No	No	No	During surgery (18 patient with US findings suggestive of LUS rupture and dehiscence)	
Edwards D. et al. [23]	2023	35786305	No	No	No	10/3	
Eleje G.U. et al. [24]	2023	37009324	No	No	No	38 2/7 during surgery	
Mutiso SK. et al. [25]	2024	38183151	Yes	No	No	11 0/7	
Zhang J. et al. [26]	2014	25864270	No	No	No	38 1/7 during surgery	
Wye D. et al. [27]	2014	28191206	No	No	No	30 0/7	
Hamar B.D. et al. [28]	2003	14607034	No	No	No	20 0/7	
Zhu Y. et al. [17]	2021	34309465	No	No	No	24 0/7-37 0/7	
Fox NS. et al. [30]	2014	24785605	20	No	No	During surgery	
Tischner I. et al. [30]	2010	20226423	Yes	No	No	27 0/7	
Filipcikovaa R. et al. [31]	2014	23446209	No	No	No	Before pregnancy	
Sbarra M.et al. [32]	2009	19482275	No	No	No	During surgery	
Balachandran Nair D. et al. [33]	2016	27358091	No	No	No	During surgery	
Inovay J. et al. [34]	1999	10527971	No	Yes	No	15	
Zuckerwise L.C. et al. [35]	2011	21768866	No	No	No	19	
**Author**	**Year**	**PMID**	**Gestational weeks at birth**	**Fetal demise**	**US** **images**	**RMN images**	**Surgery images**
Zermano S. et al.	2024		29 1/7	No	Yes	Yes	Yes
Cui X. et al. [22]	2020	32756164	>37 (64 patients) <37 (43 patients)	No	No	No	No
Edwards D. et al. [23]	2023	35786305	>37	No	Yes	No	No
Eleje G.U. et al. [24]	2023	37009324	38 2/7	No	No	No	Yes
Mutiso SK. et al. [25]	2024	38183151	11 0/7	No	Yes	No	Yes
Zhang J. et al. [26]	2014	25864270	38 1/7	Yes	No	No	Yes
Wye D. et al. [27]	2014	28191206	30 0/7	Yes	Yes	No	Yes
Hamar B.D. et al. [28]	2003	14607034	31 0/7	No	Yes	Yes	No
Zhu Y. et al. [17]	2021	34309465	33 0/7–37 0/7	3 cases	Yes	No	No
Fox NS. et al. [29]	2014	24785605	34 0/7–39 0/7	No	No	No	No
Tischner I. et al. [30]	2010	20226423	27 0/7	Yes	No	No	Yes
Filipcikovaa R. et al. [31]	2014	23446209	31 0/7–38 0/7	No	Yes	No	No
Sbarra M.et al. [32]	2009	19482275	38 0/7	No	No	No	No
Balachandran Nair D. et al. [33]	2016	27358091	34 5/7	No	No	No	Yes
Inovay J. et al. [34]	1999	10527971	15 0/7	Yes	No	No	No
Zuckerwise L.C. et al. [35]	2011	21768866	21 0/7	Pregnancy termination	Yes	No	No

## Data Availability

The datasets generated and/or analysed during the current study are available from the corresponding author upon reasonable request.

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
