# Peer review of "Prenatal Detection and Conservative Management of Uterine Scar Dehiscence in Patient with Previous Uterine Rupture and Multiple Surgeries—A Case Report"

_healthcare, 2024, doi:10.3390/healthcare12100988_

Round 1
Reviewer 1 Report
Comments and Suggestions for Authors
This paper entitled “Prenatal detection and conservative management of uterine scar dehiscence in patient with previous uterine rupture and multiple surgeries for severe endometriosis and adenomyosis - A Case Report” describes a case of early detection of signs of uterine dehiscence in a patient with history of uterine rupture, and aims to hypothesize a protocol for patients with history of uterine ruptures to minimize recurrence and increase the chance of conservative management.
I recommend rejecting this paper for publication in its current state
Major:
- Authors mention repeatedly the history of endometriosis/adenomyosis and the surgeries that the patient underwent as a risk factor for uterine rupture. However, what the authors fail to mention is that abdominal adhesions have been shown to be a protective factor against uterine rupture, or at least has caused delay in the diagnosis of the uterine rupture as it offers an additional layer of containment.
- Figures needs major editing: titles and legends
- The text needs major formatting as it is very disorganized.
- All abbreviations need to be explained the first time they are used in the text (MgSO4, 28w, DIE, etc)
- Line 154-156: why did the patient receive tocolysis? Did she have contractions and was showing signs of labor? Why did the patient receive MgSO4? What she in labor with risk of imminent delivery? When was all of this given (patient was in the hospital between 24w5d and 29w, please specify when she received these meds)
- Discussion needs much more elaboration, maybe discuss a protocol for uterine monitoring for patient with history of uterine rupture
- A lot of emphasis was put on VBAC in this text, although it is irrelevant to the current case
Minor
- Title : please remove « for severe endometriosis and adenomyosis”
- English language requires major editing
- Line 83: should be G2P1A0L0 as patient had a delivery of stillborn at 22 weeks (after 20 weeks is considered para)
- Line 53-56 and line 61-64: what is the relevance of this text? I would recommend removing
- Paragraph 2: change title from “Results” to “Case report”
- Line 90: please elaborate on what you mean by “compromised situation”
- Line 98: replace “sactosalpinge” with “sactosalpinx”, and please define if it was hydro, hemo or pyosalpinx.
- Line 115: replace “died” with “stillborn fetus”
- Figure 3-4: Please explain what you mean by “spongy”
Comments on the Quality of English Language
The english language in the text requires major editing.
Author Response
Reply comment 1: The new version of the paper describes the possibility that previous surgery may delay the diagnosis of uterine dehiscence due to the presence of adhesions. We noted how this led us to investigate the ultrasound findings in more detail, even though the patient was asymptomatic. Lines 191-199
Reply comment 2:We have completely modified all figures, we have also inserted a more detailed caption
Reply comment 3-4:we reformatted the text and replaced all abbreviations
Reply comment 5: We have described the patient's medical history in more detail, reporting the reason and weeks of gestation when drug therapies were administered
Reply comment 6:In the discussion, we included how we manage patients with previous uterine rupture and previous abdominal surgery in our Clinic. However, as this is a case report, we do not have enough data to draw up a specific management protocol for these patients; in fact, there are no specific guidelines on the subject and it would be necessary to investigate the topic further through further clinical studies.
Reply comment 7:We have removed the part of the text relating to VBAC.
Reply comment 8: we remove " for severe endometriosis and adenomyosis" in the title
Reply comment 9: a new author Nadia Parisi edited the English language
Reply comment 10: we add G2P1A0L0 in the text
Reply comment 11: we remove Line 53-56 and line 61-64
Reply comment 12: we change title from “Results” to “Case report”
Reply comment 13: we described more in detail what we mean for “compromised situation”
Reply comment 14: we replace “sactosalpinge” with “hemosalpinx".
Reply comment 15: we replace “died” with “stillborn fetus”
Reply comment 16: we used another term instead of “spongy”

Reviewer 2 Report
Comments and Suggestions for Authors
This manuscript appears to be a valuable and useful clinical case report. The structure of the paper is fine, but there are some areas that need to be corrected. If they are corrected, the manuscript could be considered acceptable.
The following is a list of the areas that need to be corrected.
1. p. 1 [abstract], line 24: "We present a rare case ~ for severe endometriosis and adenomyosis.": A brief summary of this case is needed after this sentence.
2. p. 3, line 98, "DIE"; p. 4, line 125, "ART"; p. 4, line 130, "MFM": full spelling is required for each.
3. p. 8, line 178, "a Pico negative pressure wound therapy": should be written as PICO™ system PICO system (Smith and Nephew Medical Ltd, Hull, UK), etc.
4. p. 9, lines 210-212, "Regarding the management of these patients, ~the timing of the caesarean section.": This sentence needs to be cited.
5. p. 9, lines 223-227, ""Regarding the management of these patients, ~the timing of the caesarean section.": This sentence needs to be cited.
6. p. 9, line 246, "adonomiosis": is this a misspelling?
7. References: Please describe how to describe references in accordance with “Instruction for Authors”.
References should be described as follows, depending on the type of work: Journal Articles:
1. Author 1, A.B.; Author 2, C.D. Title of the article. Abbreviated Journal Name Year, Volume, page range.
8. Figures: Need explanation of figures: Each part of the photo should be indicated by arrows or other means to explain the photo. In particular, Figure 2 is not at all clear what it shows. "Instructions for Authors" instructs "All Figures, Schemes and Tables should have a short explanatory title and caption.". Also, words for explanation should not be filled in the figure, but should be indicated by arrowheads, arrows, etc., and what they refer to should be explained in Figure legends.
9. Figures 5 and 6: The lesion is too small and the letters and lines obscure the findings. Enlarge the lesion and add arrows so that the lesion is not obscured.
Author Response
Reply comment 1: The new version of the abstract described more in detail the patient medical history.
Reply comment 2: we have replaced all abbreviations
Reply comment 3: we add correct name PICO™ system PICO system (Smith and Nephew Medical Ltd, Hull, UK), etc.
Reply comment 4-5: we add citation number.
Reply comment 6: we change with adenomyosis
Reply comment 7: we editing all the references in accordance with “Instruction for Authors”.
Reply comment 8-9: we modified all the figures, adding a more detailed legend and insert arrows,lines,arrowhead etc.

Reviewer 3 Report
Comments and Suggestions for Authors
The authors present a case report and brief literature review of a patient with US/MRI imaged uterine dehiscence in her current pregnancy. Her past history was significant for endometriosis, adenomyosis, multiple prior surgical procedures and a second semester uterine rupture with fetal demise. She was managed expectantly for several weeks following diagnosis of her uterine dehiscence on a routine anatomic US, but underwent c section due to intractable contractions.
Although interesting, the case report and literature review do not add anything to the existing literature. The use of imaging, including US and MRI, is well-documented. In addition to the references provided by the authors, there are multiple other reports. I've included several as examples:
To report obstetric outcomes in a series of women with prior uterine rupture or prior uterine dehiscence managed with a standardized protocol.
Patients with prior uterine rupture or uterine dehiscence were followed with serial ultrasound scans to assess fetal growth and lower uterine segment integrity.
Patients with prior uterine rupture or uterine dehiscence can have excellent outcomes in subsequent pregnancies if managed in a standardized manner, including cesarean delivery before the onset of labor or immediately at the onset of spontaneous preterm labor.
Fox NS, Gerber RS, Mourad M, Saltzman DH, Klauser CK, Gupta S, Rebarber A. Pregnancy outcomes in patients with prior uterine rupture or dehiscence. Obstet Gynecol. 2014 Apr;123(4):785-9. doi: 10.1097/AOG.0000000000000181. PMID: 24785605.
Uterine dehiscence and rupture are serious complications of pregnancy after a cesarean delivery. Management of uterine dehiscence diagnosed in second trimester can be controversial.
With close monitoring, expectant management of uterine dehiscence diagnosed in the second trimester is possible.
Hamar BD, Levine D, Katz NL, Lim KH. Expectant management of uterine dehiscence in the second trimester of pregnancy. Obstet Gynecol. 2003 Nov;102(5 Pt 2):1139-42. doi: 10.1016/s0029-7844(03)00162-5. PMID: 14607034.
Imaging can assist in diagnosis of both acute and chronic complications of CD. The choice of imaging modality usually depends on the type and acuity of symptoms, severity of symptoms, and the anatomic site of interest. Symptoms, including heavy bleeding, fever, and other clinical signs of postoperative infection, prompt an investigation via imaging. Ultrasound is usually the first imaging modality of choice; however, CT is used relatively liberally after ultrasound for evaluation of common and/or serious postoperative complications. MRI may be used as a problem-solving tool.
The rate of CDs continues to rise, whereas the rate of vaginal delivery after cesarean birth continues to decline. Many women now tend to undergo multiple CDs, and therefore the associated chronic maternal morbidities are of growing concern. Accurate diagnosis of these conditions is crucial in maternal and fetal well-being. Many of these complications are diagnosed by imaging, and radiologists should be aware of the type and imaging appearances of these conditions.
Moshiri M, Osman S, Bhargava P, Maximin S, Robinson TJ, Katz DS. Imaging evaluation of maternal complications associated with repeat cesarean deliveries. Radiol Clin North Am. 2014 Sep;52(5):1117-35. doi: 10.1016/j.rcl.2014.05.010. Epub 2014 Jul 3. PMID: 25173662.
Comments on the Quality of English LanguageNeeds some editing should the manuscript be accepted
Author Response
Reply comment 1: Thank you for commenting on our paper.
We are aware that there are numerous papers in the literature describing clinical cases with previous uterine rupture, as reported in the references or added by you in the comment.
However, we believe that the strength of our work lies in the particularity of the clinical history of the patient, who had previously undergone up to six abdominal surgeries, and also in the good imaging documentation.
In the new version, we have improved the quality of the English, as requested.

Round 2
Reviewer 3 Report
Comments and Suggestions for Authors
In my previous review, I commented ""Although interesting, the case report and literature review do not add anything to the existing literature. The use of imaging, including US and MRI, is well-documented."
The authors replied, “We are aware that there are numerous papers in the literature describing clinical cases with previous uterine rupture, as reported in the references or added by you in the comment. However, we believe that the strength of our work lies in the particularity of the clinical history of the patient, who had previously undergone up to six abdominal surgeries, and also in the good imaging documentation.”
However, the authors fail to provide a compelling argument. Undergoing multiple prior surgeries is a well-known risk factor for uterine rupture. There are innumerable papers, books and on-line images demonstrating endometriosis, uterine dehiscence, uterine rupture, and intraoperative findings.
Comments on the Quality of English LanguageMinor editing
Author Response
Thank you for your comment. We have expanded the literature review in the Conclusion section and included a summary table.
